# Cytoplasmic NOTCH and membrane-derived β-catenin link cell fate choice to epithelial-mesenchymal transition during myogenesis

Daniel Sieiro[1,2†], Anne C Rios[1†‡], Claire E Hirst[1], Christophe Marcelle[1,2*]

[1]Australian Regenerative Medicine Institute, Monash University, Clayton, Australia; [2]Institut NeuroMyoGene, University Lyon 1, CNRS UMR 5310, INSERM U 1217, Villeurbanne, France

**Abstract** How cells in the embryo coordinate epithelial plasticity with cell fate decision in a fast changing cellular environment is largely unknown. In chick embryos, skeletal muscle formation is initiated by migrating Delta1-expressing neural crest cells that trigger NOTCH signaling and myogenesis in selected epithelial somite progenitor cells, which rapidly translocate into the nascent muscle to differentiate. Here, we uncovered at the heart of this response a signaling module encompassing NOTCH, GSK-3β, SNAI1 and β-catenin. Independent of its transcriptional function, NOTCH profoundly inhibits GSK-3β activity. As a result SNAI1 is stabilized, triggering an epithelial to mesenchymal transition. This allows the recruitment of β-catenin from the membrane, which acts as a transcriptional co-factor to activate myogenesis, independently of WNT ligand. Our results intimately associate the initiation of myogenesis to a change in cell adhesion and may reveal a general principle for coupling cell fate changes to EMT in many developmental and pathological processes.

**\*For correspondence:** christophe. marcelle@monash.edu

[†]These authors contributed equally to this work

**Present address:** [‡]Walter and Eliza Hall Institute of Biomedical Research, Melbourne, Australia

**Competing interests:** The authors declare that no competing interests exist.

## Introduction

During early embryogenesis, a succession of extensive tissue rearrangements and cell migration events, intimately associated with rapid cell fate changes, establish the tissues and organs of the future adult. A model where such complex issues are amenable to experimentation is the early formation of skeletal muscles in the chick embryo.

Over many days of development, the medial border of the dermomyotome (DML) generates the first skeletal muscle cells that assemble into a primary myotome (*Figure 1A–D*). This arises from a crucial cell fate decision: epithelial cells in the DML either self-renew or undergo myogenic differentiation and this is accompanied by an EMT that allows their translocation into the primary myotome (*Denetclaw et al., 1997*; *Gros et al., 2009*; *Ordahl et al., 2001*; *Rios et al., 2011*).

Classical experiments demonstrated that signaling cues from tissues surrounding the somites act as inducers of muscle formation (*Ordahl and Le Douarin, 1992*). It was later shown that WNT1 and WNT3a are expressed in the dorsal neural tube and that, in vitro, both ligands enhance myogenesis in somites (*Munsterberg et al., 1995*; *Stern et al., 1995*; *Tajbakhsh et al., 1998*). A major consequence of the activation of the 'canonical' WNT pathway by WNT ligands is that β-catenin, normally degraded by the APC/Axin destruction complex, accumulates in the cytoplasm and enters the nucleus, where it engages DNA-bound TCF transcription factor to activate their targets (hence the name of WNT/β-catenin-dependent pathway to this cellular response). Since it was demonstrated that MYF5 promoter contains TCF binding sites required for its in vivo expression in the DML

(*Borello et al., 2006*) and that dominant negative and constitutively active forms of TCF and β-catenin modulate MYF5 expression in somites (*Abu-Elmagd et al., 2010*; *Gros et al., 2009*), this has led to the largely accepted theory that myogenesis in somites is under the control of WNTs from the dorsal neural tube acting through a WNT/β-catenin-dependent pathway.

This raises a paradox: although presumably all cells in the DML are equally exposed to WNTs from axial structures and are fully competent to initiate myogenesis, only a small proportion of DML cells do so at any given time (*Rios et al., 2011*), suggesting that other mechanisms affect the cell fate decision in selected DML cells.

We recently introduced NOTCH signaling as a novel player in early myogenesis, by demonstrating that the activation of MYF5 and MYOD in the DML is dependent upon the transient activation of NOTCH signaling, triggered by Delta1-positive neural crest cells migrating from the dorsal neural tube (*Rios et al., 2011*). This finding is compatible with the well-documented role of the dorsal neural tube in myogenesis. Importantly, the mosaic expression of Delta1 in the migrating neural crest cell population ensures that NOTCH signaling is regularly triggered in selected DML cells, thus explaining the cell fate choice necessary to generate myotomal cells over an extended period of time, all while self-renewing the progenitor population. However, this finding was not in accordance with the established role of WNT in myogenesis, nor did it explain why the initiation of the myogenic program in selected progenitors within the DML is associated with their translocation into the primary myotome *via* an EMT.

Here, we discovered that in DML cells, the activation of NOTCH signaling by Delta1-positive neural crest cells strongly decreases GSK-3β activity, independent of NOTCH transcriptional function. This leads to a dramatic stabilization of SNAI1, resulting in the initiation of the EMT program. As a consequence, β-catenin from the cell membrane pool is mobilized, allowing its entry into the nucleus where it activates MYF5 expression in a WNT ligand-independent manner. Therefore, our results suggest that non-canonical functions of NOTCH and β-catenin, associated in an efficient signaling circuitry, explain the coupling of myogenesis with changes in cell adhesion in the DML.

## Results

### Co-activation of NOTCH and WNT reporters in early myogenesis

The first sign that myogenesis is initiated in the DML is the activation of MYF5 (or MYOD) expression, which serves as a read-out of a cell fate change in this structure (*Rios et al., 2011*). As WNT and Delta1 are reported to activate myogenesis (*Munsterberg et al., 1995*; *Rios et al., 2011*; *Stern et al., 1995*; *Tajbakhsh et al., 1998*), it was important to determine whether and how these pathways co-operate in this process. As a first step, we co-electroporated the DML of trunk-level somites (as shown in *Figure 1A–D*) with reporter constructs for the NOTCH and WNT pathways. The NOTCH reporter (*Ohtsuka et al., 2006*) contains the HES1 promoter region upstream of a destabilized red fluorescent protein. HES1 is a direct target of the NOTCH pathway, and we previously showed this construct serves as a faithful reporter of NOTCH activity in somites (*Rios et al., 2011*). The 'TOPflash' reporter (*Korinek et al., 1997*; *Rios et al., 2010*) contains twelve TCF-1 binding sites upstream of a destabilized EGFP (12Tf-d2EGFP). The activity of the TOPflash reporter is triggered by the binding of the transcription factor TCF/LEF together with its co-factor β-catenin. The destabilized fluorescent reporter proteins present in both constructs allow the visualization of only the cells that are actively engaged in NOTCH and WNT signaling, while the intensity of the signal indicates the magnitude of the response.

We found that, within the DML, almost half of the epithelial cells that activated the TOPflash reporter were also positive for the NOTCH reporter (*Figure 2A,B*). Remarkably, nearly all DML cells that were positive for both reporters were MYF5-positive (*Figure 2C*), indicating a strong correlation between the activation of NOTCH and WNT reporters and the initiation of the myogenic program.

During this analysis, we made the surprising observation that a large majority of NOTCH reporter-positive cells displayed a high TOPflash fluorescence. On the contrary, most cells that expressed low levels of TOPflash fluorescence were NOTCH-reporter negative (*Figure 2A,D*). It was important to show that this difference was not due to electroporation, since DNA transfer with this technique is inherently variable between cells. To show that the variable levels of TOPflash activity were not the result of variable amounts of electroporated plasmids, we plotted the intensity of

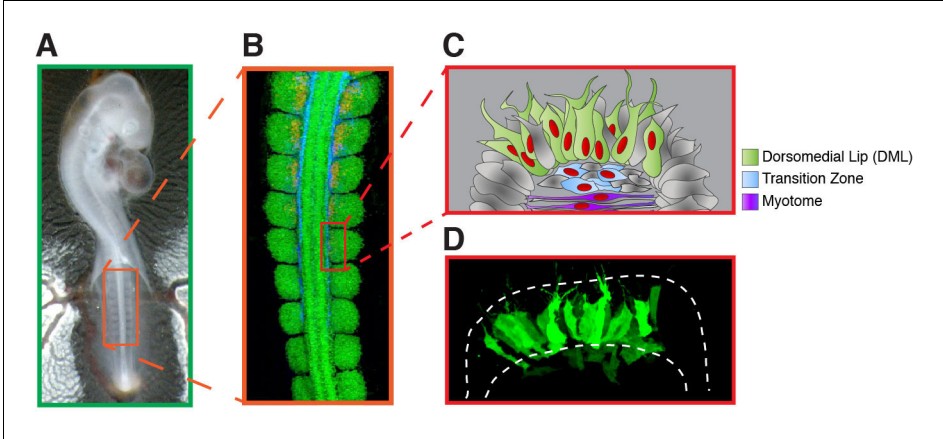

**Figure 1.** Summary of the experiments performed in this study. (**A**) Chick embryos at HH stage 15–16 (24–28 somites) were electroporated in the medial part of the (4–5) newly formed somites (boxed region). (**B**) Immunostaining of a HH 15 chick embryo with PAX7 (in green), HNK1 (in blue) and MYF5 (in red) to label the dermomyotome, the neural crest and the myotome, respectively. (**C**) Schematic illustrating regions of the somite that are represented in the confocal stacks of images shown throughout the study. It shows the medial portion of a somite 6–24 hr after its DML was electroporated. Typically, it leads to the mosaic expression (in green) of the electroporated construct(s) in the DML, the transition zone (in blue) and the nascent primary myotome (in purple). (**D**) is a maximum intensity projection of a confocal stack of a somite electroporated with GFP 6 hr prior to analysis.

TOPflash response against that of the BFP as a control. BFP is driven by an ubiquitous promoter and its fluorescence should thus be directly correlated with the quantity of plasmid incorporated in each cell. We found no correlation between the intensity of the TOPflash reporter and that of BFP (*Figure 2E*). This suggests that the low and high levels of TOPflash fluorescence observed in DML cells are due to genuine differences in TCF/β-catenin transcriptional activity. The graph also distinguishes cells that are positive (in red) for the NOTCH reporter from those that are negative (in blue, *Figure 2E*). It confirms that the vast majority of NOTCH reporter-positive (and MYF5-positive) cells display a high TCF/β-catenin transcriptional activity. This is the cell population that will be thoroughly analyzed in this study. The identity of the cell population that displays a low TCF/β-catenin transcriptional activity (about 60% of the entire TOPflash positive cell population) but is NOTCH reporter- and MYF5-negative will become clear in the following chapters.

## NOTCH signaling regulates TCF/β-catenin transcriptional activity during myogenesis

The results above raised the possibility that NOTCH and the TCF/β-catenin transcriptional activity may be interconnected during the initial phases of myogenesis. To address this, we first co-electroporated the DML with a constitutively active form of β-catenin (*Rios et al., 2010*) together with the NOTCH reporter. This did not lead to any change in activity of the NOTCH reporter, compared to controls (*Figure 3A–C*). In contrast, the co-expression of a constitutively active form of NOTCH1 (NICD [*Daudet and Lewis, 2005*]) led to rapid and robust increase in the activity of the TOPflash reporter, compared to controls (*Figure 3D,E*). To further investigate the interrelation between both pathways, we used a dominant-negative form of the NOTCH co-activator Mastermind (DN-MAML1 [*Weng et al., 2003*]). We previously showed that it strongly inhibits MYF5 expression in the DML (*Rios et al., 2011*). A constitutively active form of the most downstream effector of the WNT pathway, LEF1/TCF1 (named CA-LEF1 [*Abu-Elmagd et al., 2010*]), co-electroporated with DN-MAML1 not only rescued the inhibition of MYF5 expression by DN-MAML1, but also led to a strong activation of its expression that was indistinguishable from that observed after electroporation of CA-LEF1 alone (*Figure 3F,G*) or NICD (*Rios et al., 2011*). These data support the premise that NOTCH regulates TCF/β-catenin transcriptional activity in the DML.

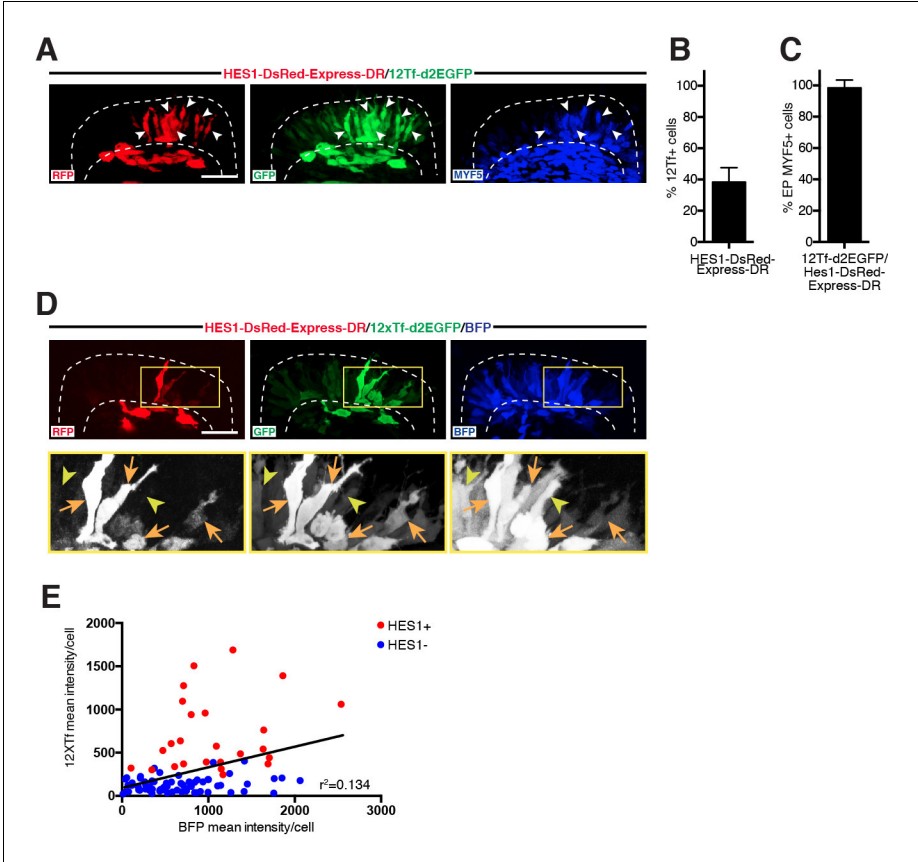

**Figure 2.** NOTCH signaling is associated with elevated TOPflash activity in early myogenesis. (**A,D**) Confocal stacks of somites, in dorsal view, 12 hr after electroporation of HES1-DsRedExpress-DR (NOTCH reporter, in red), 12Tf-d2EGFP (TOPflash reporter, in green), an ubiquitously expressed H2B-BFP (blue in D) or immunostained for MYF5 expression (blue in A). (**B–C**) Bar charts showing (**B**) 38% of cells activating the NOTCH reporter within the TOPflash reporter-positive population and (**C**) 98.5% of cells expressing MYF5 that co-activate the NOTCH and the TOPflash reporters. Lower panels (black and white in D) show enlargements of the yellow boxes (upper panels in D). Orange arrows indicate DML cells positive for the NOTCH reporter and strongly positive for the TOPflash reporter; yellow arrowheads show cells negative for NOTCH and weak or strong for TOPflash. (**E**) Linear regression chart plotting mean fluorescence intensity of CAGGS-BFP-positive cells against the mean fluorescence intensity of 12Tf-d2EGFP-positive cells ('Y=0.2378*X+93.61', $r$(96)=0.134, p<0.0002). Indicated is best-fit regression line for entire population (Hes1+: NOTCH reporter-positive, red dots; Hes1-: NOTCH reporter-negative, blue dots) and r-squared value. In each panel are indicated the antigens that were detected by immunostaining. Abbreviation: EP: electroporation. Scale bars: 50 µm

## The activation of TCF/β-catenin transcriptional activity by NOTCH is WNT ligand-independent

The TOPflash reporter reflects TCF/β-catenin transcriptional activity and it is a known outcome of WNT ligand binding to its cognate receptor. However, the results above indicate that NOTCH somehow regulates TOPflash activity in the DML. To understand this apparent contradiction, we decided to test whether WNT ligands are at all required during this process. *Dickkopf* (*Dkk*) genes comprise an evolutionarily conserved family that encode secreted proteins that antagonize WNTs, by inhibiting in the extracellular space the WNT co-receptors Lrp5 and 6 (*Niehrs, 2006*). First, we tested whether DKK1 efficiently inhibits WNT signaling by co-electroporating DKK1 into the neural tube of developing embryos, together with the TOPflash reporter described above. As control, the TOPflash reporter was electroporated alone. Without DKK1, we observed a robust TOPflash reporter activity in the neural tube (*Figure 4—figure supplement 1A,B*). The co-electroporation of DKK1 abrogated nearly all reporter activity in this tissue (*Figure 4—figure supplement 1C*). These results suggest

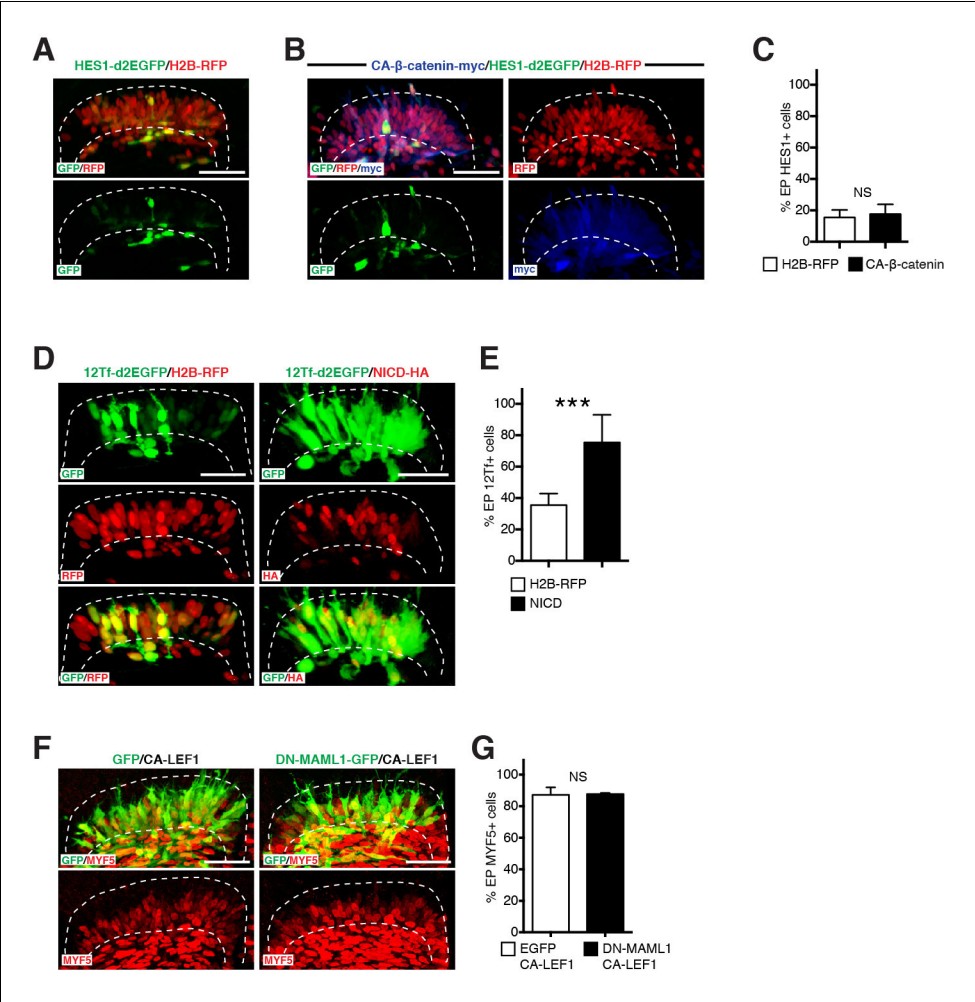

**Figure 3.** NOTCH regulates regulates TCF/β-catenin transcriptional activity during myogenesis. (**A,B**) Confocal stacks, 12 hr after electroporation of (**A**) the NOTCH reporter HES1-d2EGFP in green, H2B-RFP (in red) and (**B**) with a constitutively active form of β-catenin (in blue). (**C**) Bar charts showing 15.6% of HES1-d2EGFP-positive cells in the control (in white) or 17.7% with CA β-catenin (in black). (**D**) Confocal stacks of somites 6 hr after co-electroporation of 12Tfd2EGFP (TOPflash reporter, in green), and an ubiquitously expressed H2B-RFP (in red, left) or NICD (in red, right). (**E**) Bar charts showing 35.5% of cells expressing the TOPflash reporter in the controls (in white) or 75.4% with NICD (in black). (**F**) Confocal stacks of somites 6 hr after co-electroporation of a constitutive active form of LEF1 and GFP (in green, left) or a dominant negative form of MAML1 fused to GFP (in green, right) and a constitutive active form of LEF1; MYF5 expression is shown in red. (**G**) Bar charts showing 87.2% of MYF5-positive cells in DN-MAML1/CA-LEF1-positive cells and 87.8% MYF5-positive cells when expressing CA-LEF1 alone. In each panel are indicated the antigens that were detected by immunostaining. Abbreviation: EP: electroporation. Scale bars: 50 μm

that DKK1 efficiently inhibits most, if not all TOPflash (i.e. TCF/β-catenin transcriptional) activity in the posterior neural tube by antagonizing WNTs in the extracellular space.

We then performed a similar experiment in somites. A striking difference from the neural tube control experiment was that, even in the presence of DKK1, the TOPflash reporter was still active (*Figure 4A,B*). While this response showed a decrease of about half in the proportion of TOPflash reporter-positive cells compared to controls (*Figure 4C*), remarkably nearly all cells that remained TOPflash-positive also expressed MYF5 (*Figure 4D*). As a consequence, the overall proportion of MYF5-positive cells was unaffected (*Figure 4E*). Of these cells that remained TOPflash-positive in the presence of DKK1, nearly all expressed the NOTCH reporter (*Figure 4F,G*), a result coherent with our observation that MYF5 is tightly associated with the activation of this reporter (*Figure 2C*).

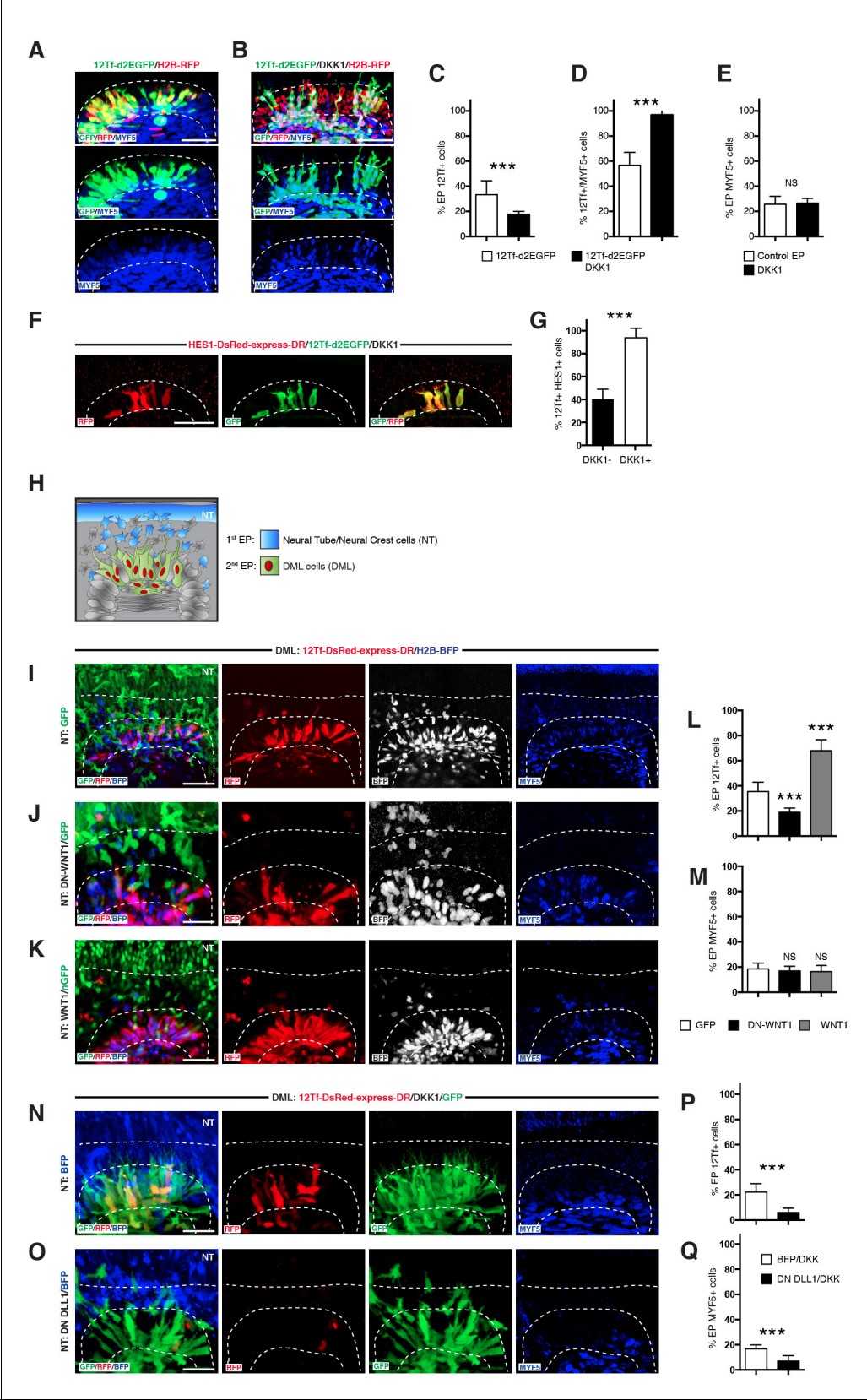

**Figure 4.** Activation of TCF/β-catenin transcriptional activity and myogenesis by NOTCH is WNT-ligand independent. (**A,B**) Confocal stacks of somites, 6 hr after co-electroporation of 12Tf-d2EGFP (TOPflash reporter, in green), with H2B-RFP (in red) without (**A**) or with (**B**) DKK-1, and immunostained for
*Figure 4 continued on next page*

Figure 4 continued

MYF5 (in blue). (C–E) Bar charts showing (C) 33.3% of TOPflash-positive cells in the controls (in white) or 17.7% with DKK-1 (in black); (D) Bar charts indicating 56.8% of TOPflash-positive cells are MYF5-positive in control embryos, while 96.9% of TOPflash-positive cells are MYF5-positive when DKK1 is present; (E) percentage of electroporated cells that are MYF5-positive in controls: 25.7%; with DKK1: 26.6%. (F) Confocal stacks of somites after co-electroporation of HES1-DsRed-express-DR (NOTCH reporter, in red) or with 12Tf-d2EGFP (TOPflash reporter in green) and DKK1. (G) Bar charts showing the% of electroporated cells co-expressing the TOPflash and the NOTCH reporters when DKK1 is present (91.4%, in black) or absent (39.9%, in white). (H) Schematic representing the design of the double electroporation experiments, with a first electroporation targeting the neural tube (NT) and the migrating neural crest (in blue), followed by electroporation in the DML. (I–K) Confocal stacks of somites showing the expression of 12Tf-DsRed-express-DR (TOPflash reporter, in red) and the electroporation marker H2B-BFP (in blue) in the DML. Dorsal neural tube was electroporated with (I) GFP, (J) dominant-negative WNT1 (in green), or (K) wild-type WNT1 (in green). (L) Bar charts showing 19% of DN-WNT1-positive electroporated cells express the TOPflash reporter (in black), 67.6% of WT-WNT1-positive electroporated cells express the TOPflash reporter (in gray), controls (in white): 35.5%. (M) Bar charts showing 15.04% of DN-WNT1-positive electroporated cells express MYF5 (in black), 15.07% for WT-WNT1-positive electroporated cells (in gray), controls (in white): 15.08%. (N,O) Confocal stacks of somites showing the TOPflash reporter 12Tf-DsRed-express-DR (in red), the electroporation marker GFP (in green) and DKK1, electroporated into the DML. The dorsal neural tube was electroporated with (N) BFP or (O) dominant-negative DLL1 and BFP. (P) Bar charts showing the% of electroporated cells expressing the TOPflash reporter, DN-DLL1 + DKK1: 6%, in black; controls with DKK1: 22.3%, in white. (Q) Bar charts showing the% of electroporated cells expressing MYF5, DN-DLL1 + DKK1: 6.8% in black; controls with DKK1: 16.6%, in white. Dotted lines indicate the borders of the neural tube and somites. Abbreviations: NT: neural tube; DML: Dorso-Medial Lip; EP: electroporation. In each panel are indicated the antigens that were detected by immunostaining with the exception of native BFP blue fluorescence. Scale bars: 50 μm.

The following figure supplement is available for figure 4:

**Figure supplement 1.** Expression of DKK1 in the neural tube abrogates WNT/β-catenin response.

These surprising results suggested that WNT ligands are dispensable for the NOTCH-dependent myogenic response in DML cells. To confirm this, we performed double electroporations (*Figure 4H*). First, we electroporated WNT1 or a dominant-negative form of WNT1 into the dorsal neural tube and the neural crest. WNT1 is predicted to activate WNT1 and WNT3a cognate receptors, while DN WNT1 is believed to act as a competitive inhibitor for both WNT1 and WNT3a (*Hoppler et al., 1996*). Second, we electroporated the TOPflash reporter and an electroporation control plasmid in somites. This allowed us to quantify the activity of the TOPflash reporter together with the myogenic response (determined by immunostaining for MYF5). We observed that WNT1 and DN WNT1 significantly increased or decreased, respectively, the activity of the TOPflash reporter, compared to controls (*Figure 4I–L*), but importantly with no visible change to their myogenic response (*Figure 4M*).

These results led us to hypothesize that the overall TCF/β-catenin transcriptional activity observed in DML cells is the sum of two TCF/β-catenin transcriptional activities: the first is NOTCH-dependent, WNT ligand-independent and it is associated with myogenesis; the second is WNT ligand-dependent, but it is not associated with myogenesis. To show this, we inhibited both DLL1 in neural crest and the WNT ligands in the DML, while monitoring the activity of the TOPflash reporter and myogenesis in somites. Compared to controls (no inhibition of DLL1 in the neural crest), this led to a near complete absence of TOPflash reporter activity in DML cells (*Figure 4O,P*), which was now accompanied by a robust decrease in their myogenic response (*Figure 4O,Q*).

Together with the results presented in *Figure 2*, these data suggest that two distinct populations coexist in the DML: the first displays a strong TCF/β-catenin transcriptional activity triggered by DLL1 from migrating neural crest cells, which results in the initiation of MYF5 expression. The second shows a low TOPflash reporter activity, initiated by WNT ligand transported by neural crest cells but with no sign of myogenesis. These findings prompted a search for a mechanism that could explain how DLL1 can trigger the high TCF/β-catenin transcriptional activity that is required for the initiation of the myogenic program.

## Membrane-derived β-catenin is required for the NOTCH-mediated activation of MYF5

In the DML, the initiation of myogenesis is accompanied by an EMT (*Gros et al., 2009*; *Rios et al., 2011*). Because it integrates two distinct functions i) as a transcriptional co-factor together with TCF/LEF and ii) as a structural adaptor protein linking cadherins to the actin cytoskeleton in cell-cell

adhesion, β-catenin is a prime candidate to play a central role in these two processes. However, despite experimental evidence that the dissociation of adherens junction (AJ) can induce the release of β-catenin into the cytoplasm, a functional connection between the nuclear and membranal pools of β-catenin in vivo is still a matter of debate (*Brembeck et al., 2006*; *Gavard and Mège, 2012*; *Nelson and Nusse, 2004*).

To test whether β-catenin located at the AJ and at the plasma membrane of DML cells is required for the activation of MYF5, we electroporated DML cells with a mutant Y489F β-catenin that cannot be mobilized from the junctional complex (*Rhee et al., 2007*). As control, wild-type (WT) β-catenin was utilized. We verified that the overall levels of exogenously provided WT and mutant β-catenin were similar to those of the endogenous β-catenin (*Figure 5—figure supplement 1*). In control embryos, we observed a normal proportion of MYF5-positive DML cells (*Figure 5A,C*). In contrast, the electroporation of Y489F β-catenin resulted in a robust reduction of MYF5 expression (*Figure 5B,C*), suggesting that junctional β-catenin is required for the early activation of myogenesis.

We then tested whether junctional β-catenin is required, downstream of NOTCH, to induce MYF5. We repeated the same experiment as above, this time in combination with NICD. Y489F or WT β-catenin ωερε expressed for 10 hr in the DML, after which (using a Tet-on inducible system, see Materials and methods) NICD expression was induced for 6 hr (*Figure 5D*). In control embryos, the induction of NICD expression resulted in a robust increase in overall (nuclear and cytoplasmic) β-catenin protein levels in electroporated cells (as determined by immunostaining for the Myc tag present on the exogenous WT β-catenin as well as by staining for endogenous β-catenin [*Figure 5E* and *Figure 5—figure supplement 2*]). In these embryos, the vast majority of NICD+/WT β-catenin+ cells expressed MYF5 as expected (*Figure 5E,G*). The myogenic response in embryos electroporated with Y489F β-catenin was strikingly different, and the presence of the mutant form of β-catenin not only prevented the activation of myogenesis by NICD, but also resulted in a dramatic reduction of MYF5 expression (*Figure 5F,G*) that was similar to the decrease observed when Y489F β-catenin was electroporated alone (*Figure 5C*). As reported (*Rhee et al., 2007*), Y489F β-catenin remained mainly distributed at the membrane and AJ of DML cells (*Figure 5—figure supplement 2*), suggesting that it did not mobilize from the junctional complex.

Together with the observation that MYF5 expression is associated with high activity of the TOP-flash reporter in DML cells (*Figures 2* and *4*), our data suggest that the pool of β-catenin at the AJ and along the plasma membrane of DML cells acts as a reservoir for the NOTCH-mediated, strong β-catenin-dependent response required for the activation of MYF5.

## SNAI1 is a necessary and sufficient step for the NOTCH-dependent activation of MYF5

Major players in cell adhesion are the zinc finger transcriptional repressors *Snail* (or SNAI) family members. An essential function of SNAI is to repress epithelial gene expression and thereby promote epithelial-mesenchymal transitions (EMT) during vertebrate and invertebrate development and metastatic progression of cancers (*Barrallo-Gimeno and Nieto, 2005*). In early chick embryos, SNAI1 is expressed in the epithelial dermomyotome, including the DML. Its activation leads to the down-regulation of N-cadherin expression present at the adherens junction of epithelial cells located in the central dermomyotome and to their subsequent EMT (*Delfini et al., 2009*). Coherent with this, the over-expression of SNAI1 in the DML induced a significant increase in the translocation of DML cells into the myotome (*Figure 6A,B*), while the inhibition of its function by a SNAI1-specific siRNA (*Delfini et al., 2009*) inhibited their translocation (*Figure 6C,D*). Since we showed above that the release of β-catenin from the AJ is a key event in the activation of MYF5, we tested whether SNAI1 could regulate the activation of MYF5 expression in the DML. The electroporation of SNAI1 in the DML induced a strong activation of MYF5 expression, compared to controls (*Figure 6F,G*). Conversely, the electroporation of a dominant-negative form of SNAIL-1 or an siRNA directed against SNAI1 (*Delfini et al., 2009*) resulted in a significant decrease in MYF5 expression (*Figure 6C,E,F,G*). Finally, we co-electroporated NICD together with the dominant-negative form of SNAI1 and observed that the massive increase in MYF5 expression observed after NICD expression (*Rios et al., 2011*) was profoundly reduced by DN-SNAI1 (*Figure 6F,G*), thus demonstrating that SNAI1 activation is a necessary step, downstream of NOTCH, in the chain of events that leads to the activation of MYF5 in DML cells.

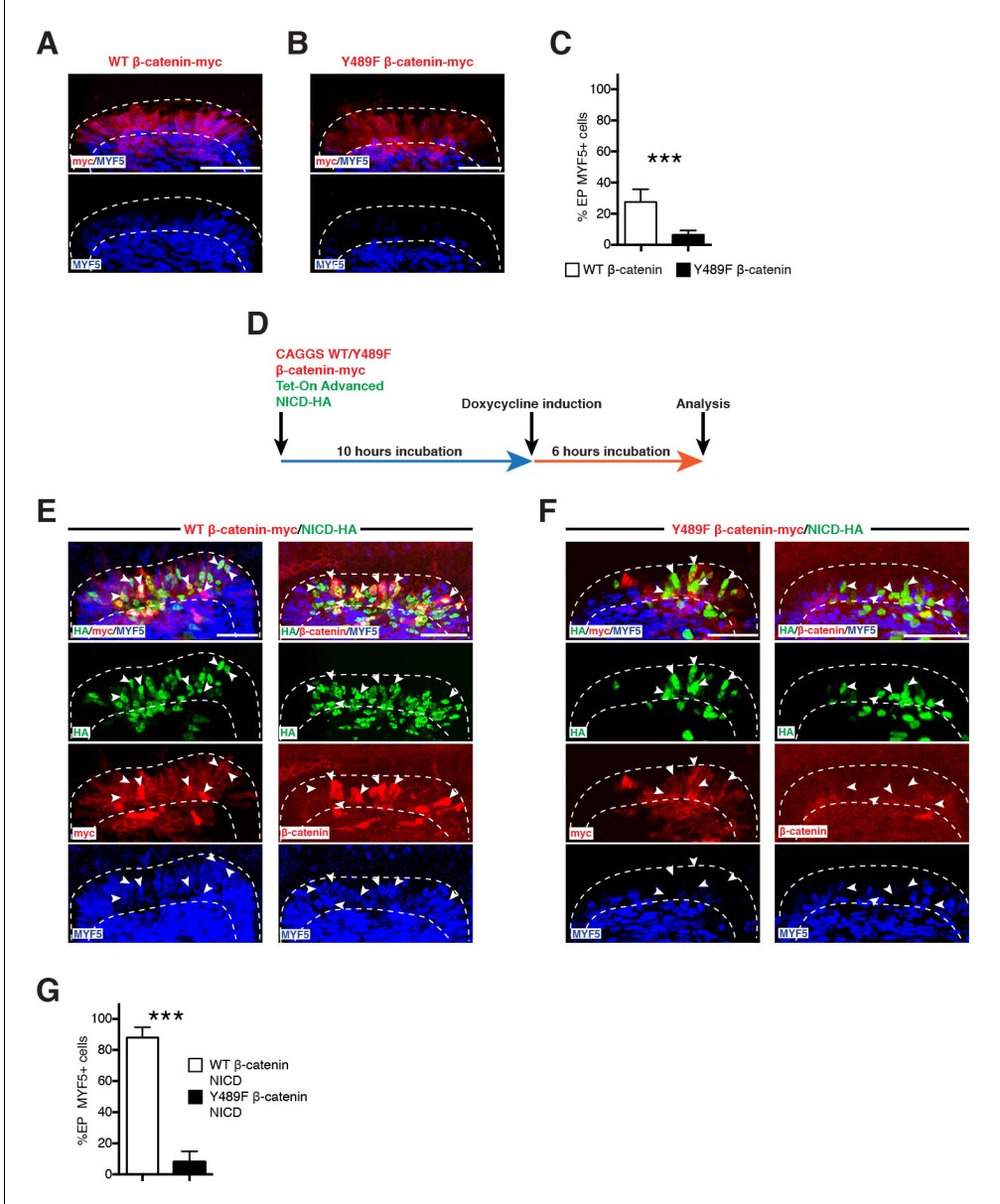

**Figure 5.** β-catenin from the cell membrane and AJ is required for the NOTCH-mediated activation of MYF5 expression. (**A,B**) Confocal stacks of somites after electroporation of WT β-catenin (A) or Y489F β-catenin (B). MYF5 expression is shown (in blue). (**C**) Bar charts showing the% of electroporated cells that were positive for MYF5 after electroporation of WT β-catenin (27.7%, in white) or Y489F β-catenin (6.4%, in black). (**D**) Schematic of the experimental design that was followed: (Myc-tagged) WT or a mutant Y489F β-catenin under a CAGGS ubiquitous promoter was electroporated in the DML together with the Tet-On Advanced transactivator (rtTA) and NICD inserted in the response vector. The β-catenin (WT or mutant) was expressed from the start of the experiment; NICD was induced 10 hr later. (**E,F**) Confocal stacks of somites after electroporation of (**E**) NICD (in green) and WT β-catenin (in red) or (**F**) NICD (in green) and Y489F β-catenin (in red). MYF5 expression is shown (in blue). (**G**) Bar charts showing the% of MYF5-positive cells after electroporation of NICD with WT β-catenin (88%, in white) or Y489F β-catenin (8.1%, in black). In each panel are indicated the antigens that were detected by immunostaining. DML: dorso-medial lip. Scale bars: 50 μm

The following figure supplements are available for figure 5:

**Figure supplement 1.** Endogenous and exogenous β-catenin are similarly expressed in the DML.

**Figure supplement 2.** Cellular localization of WT and Y489F β-catenin along with NICD.

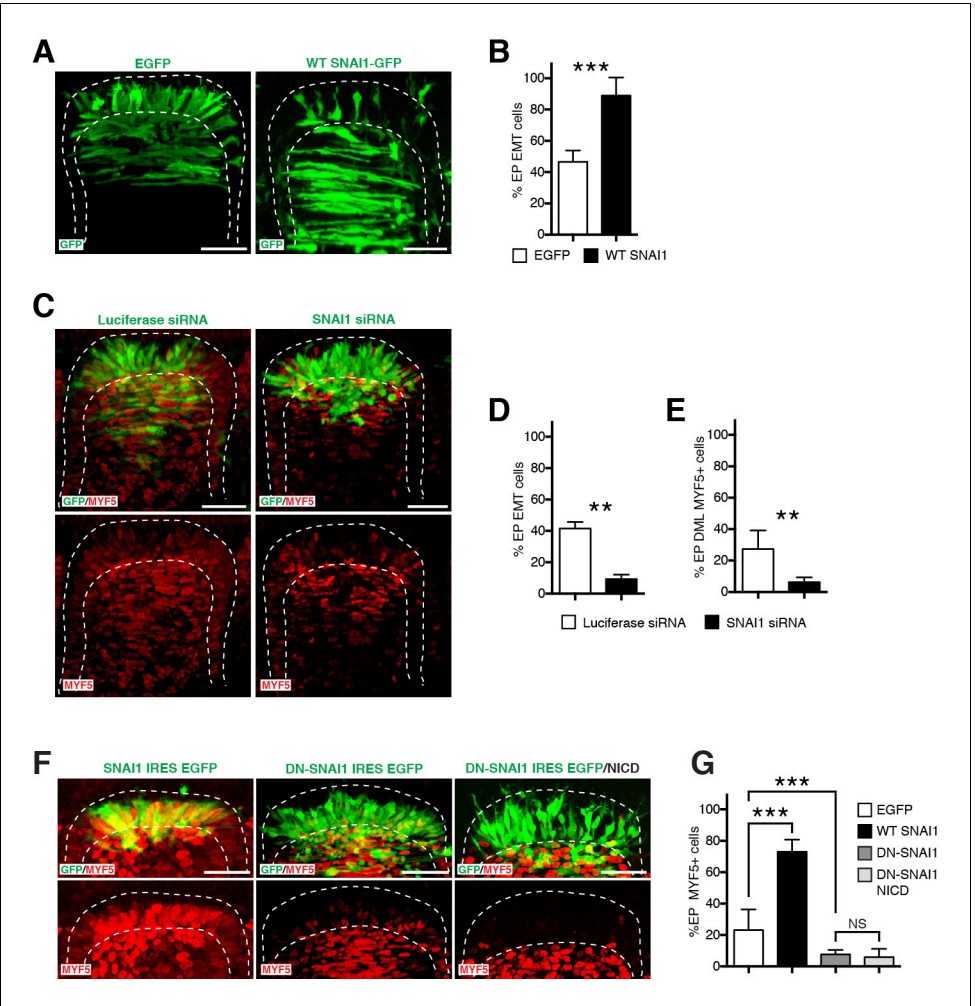

**Figure 6.** SNAI1 is a necessary and sufficient step for the NOTCH-dependent activation of MYF5. (**A**) Confocal stacks of somites, 17 hr after electroporation of EGFP as control or with SNAI1. (**B**) Bar charts showing the% of cells that have entered the primary myotome in the control (46.7%, in white) or after electroporation of SNAI1 (89%, in black). (**C**) Confocal stacks of somites, 17 hr after electroporation of a luciferase-specific siRNA as control or a SNAI1-specific siRNA, immunostained for MYF5 (in red). (**D–E**) Bar charts showing the% of electroporated cells that have entered the primary myotome (**D**) in the control (41.6%, in white) of with siRNA SNAI1 (9.3%, in black) and the% of electroporated cells that are MYF5-positive (**E**) in the control (26.2%, in white) or with siRNA SNAI1 (6.3%, in black). (**F**) Confocal stacks of somites, 6 hr after electroporation of a WT chicken SNAI1 (in green, left), a dominant negative (DN) form of SNAI1 (in green, right) alone or together with NICD. MYF5 expression is indicated (in red). (**G**) Bar charts showing the% of MYF5-positive cells after GFP electroporation (23.1%, in white), with chicken SNAI1 (73%, in black), DN SNAI1 alone (7.7%, in dark grey) or together with NICD (6%, in light grey). In each panel are indicated the antigens that were detected by immunostaining. Abbreviation: EP: electroporation. Scale bars: 50 μm.

## NOTCH regulates SNAI1 degradation through inhibition of GSK-3β activity

We then determined whether NOTCH regulates SNAI1. NICD regulates the transcription of many target genes (*Fortini, 2009*; *Guruharsha et al., 2012*), including SNAIL (*Grego-Bessa et al., 2004*). Therefore, we wondered whether this would be the case in the DML. However, the expression of NICD did not result in any significant increase in SNAI1 transcription (as judged by in situ hybridization, data not shown). These results led us to explore alternative mechanisms. SNAI1 mRNA is widely expressed; however, its activity is tightly regulated post-translationally through phosphorylation by GSK-3β, which leads to its β-Trcp-mediated ubiquitination and degradation. Due to this continuous

degradation, SNAI1 protein has a short half-life, estimated to be about 25 min (*Zhou et al., 2004*). We first determined whether NOTCH signaling could modify the stability of SNAI1 in DML cells by electroporating SNAI1 fused to GFP (SNAI1-GFP), alone or in combination with NICD. SNAI1-GFP was used in low concentration, leading to no visible phenotype, e.g. EMT or activation of MYF5, data not shown). In control embryos, GFP was undetectable under UV examination, but faintly visible after immunostaining and confocal examination in electroporated cells (*Figure 7A*). In sharp contrast, the co-electroporation of NICD with SNAI1-GFP led to a massive increase in GFP staining in most electroporated cells (*Figure 7A*), suggesting that the activation of NOTCH signaling prevents the degradation of SNAI1 that normally occurs in DML cells.

Since GSK-3β is the main regulator of SNAI1 stability (*Barrallo-Gimeno and Nieto, 2005*; *Weng et al., 2003*), we tested whether NOTCH regulates the activity of GSK-3β. Taelman and colleagues (*Taelman et al., 2010*) cleverly designed a GSK-3β fluorescent 'biosensor' (see Materials and methods). When GSK-3β is active, the biosensor fluorescence is reduced; in contrast, when GSK-3β is inactive, its fluorescence is increased. We first determined whether the GSK-3β biosensor is active in DML cells, and observed that fluorescence of the biosensor was only detected in a small proportion of electroporated DML cells (*Figure 7B*). This observation indicates that in the majority of DML cells under normal, unperturbed conditions, GSK-3β is active (i.e. low or no biosensor detectable). However, in a minority of cells, GSK-3β is inhibited (i.e. high level of biosensor). To find out the identity of the biosensor-positive cells in normal, unperturbed condition, we performed a series of co-electroporations followed by immunostaining. This showed that the cells in which the fluorescence of the biosensor is observed are those that i) activate the NOTCH reporter (*Figure 7C, D*), ii) in which SNAI1 is active (*Figure 7F,G*) and iii) are MYF5-positive (*Figure 7C,E*). This reinforces the hypothesis of a mechanistic link between the initiation of myogenesis, NOTCH activation, the stability of SNAIL and the inhibition of GSK-3β activity in DML cells.

We then tested whether NOTCH regulates GSK-3β by co-electroporating the biosensor together with NICD. This resulted in a massive increase in the fluorescence of the GSK-3β biosensor (*Figure 7B*), suggesting that NICD inhibits GSK-3β kinase activity. To eliminate the possibility that the increase in fluorescence of the biosensor was due to a non-physiological effect of NICD expression in DML cells, we induced the activation of NOTCH signaling in the DML by electroporating DLL1 in the neural crest cell population (as described in *Figure 4H*), with the GSK-3β biosensor electroporated in adjacent somites. Again, we observed a strong increase in the fluorescence of the biosensor in somites when compared to controls (*Figure 7H,I*). The electroporation of a dominant-negative form of GSK-3β (DN-GSK-3β [*Taelman et al., 2010*]) mimicked the effects of NOTCH activation on SNAI1 and MYF5, resulting in a significant increase in the proportion of DML cells undergoing EMT (*Figure 7J,K*) as well as a robust increase in MYF5 expression in DML cells (*Figure 7L*).

Altogether our data suggest that a direct consequence of the activation of NOTCH signaling by migrating DLL1-positive neural crest cells is an inhibition of GSK-3β activity that leads to a stabilization of the SNAI1 protein, which triggers an EMT of DML cells.

## NOTCH controls myogenesis independently of its transcriptional role in the nucleus

The decrease of GSK-3β activity in DML cells could be a transcriptional response to the activation of NOTCH signaling. However, NOTCH can also act in a manner independent to RBPJ/CSL co-activation (*Andersen et al., 2012*; *Ayaz and Osborne, 2014*). To distinguish between the two possibilities, we electroporated a constitutively-active and a dominant-negative form of RBPJ (*Kuroda et al., 1999*) in DML cells. As expected, the DN-RBPJ and CA-RBPJ significantly inhibited or activated the NOTCH reporter, respectively, compared to controls (*Figure 8A–D*). However, both constructs failed to modify the expression of MYF5 in this structure, compared to controls (*Figure 8E*). These results indicate that NICD regulates myogenesis in the DML independently of its transcriptional activity with RBPJ.

We hypothesized that NICD could be mediating this activity in the cytosol. To address this, we constructed an artificial, membrane-tethered, HA-tagged NICD (CD4-NICD, see Materials and methods). Unlike wild-type NICD, which upon electroporation readily enters the nucleus and massively activates the NOTCH reporter (*Rios et al., 2011*), immunostaining for CD4-NICD was not observed in the nucleus and, coherent with this, it did not activate the NOTCH reporter (*Figure 9A,B*). Despite this, CD4-NICD expression resulted in a massive stabilization of the

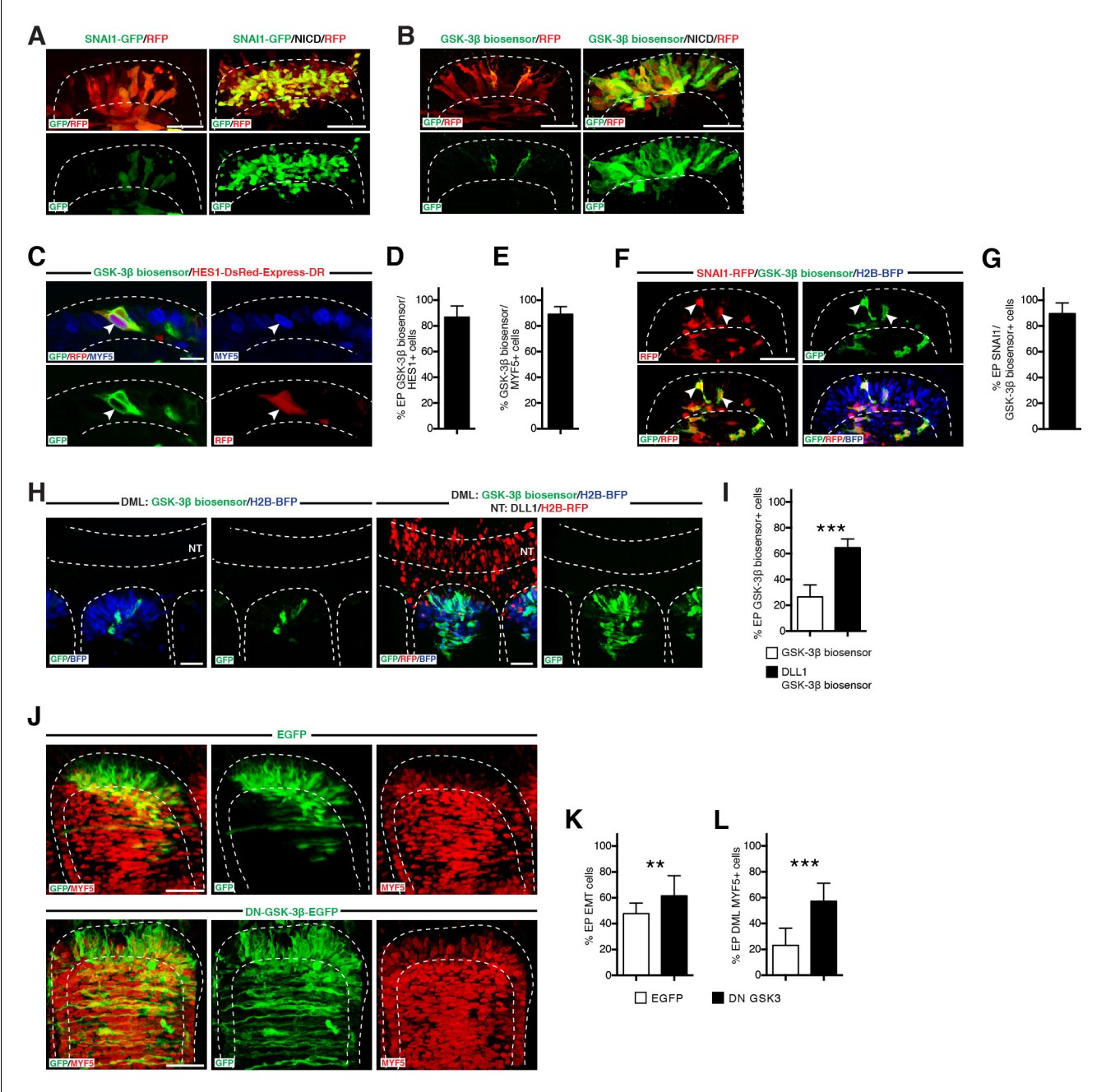

**Figure 7.** NOTCH regulates SNAI1 degradation through inhibition of GSK-3β activity. (**A**) Confocal stacks of somites, 6 hr after electroporation of RFP (in red) and SNAI1-GFP (fusion of SNAI1 and GFP, in green) alone or together with NICD. (**B**) Confocal stacks of somites, 6 hr after electroporation of GSK-3β biosensor (in green) and RFP (in red) alone (left), or together with NICD (right). (**C**) Confocal stacks of a somite 6 hr after co-electroporation of the GSK-3β biosensor (green) and the NOTCH reporter (red) and immunostained for MYF5 (blue). (**D,E**) Bar charts showing the% of GSK-3β biosensor-positive cells that activate NOTCH signaling (86.9%, **D**) and that are MYF5-positive (88.2%, **E**). (**F**) Confocal stacks of a somite 6 hr after co-electroporation of a SNAI fused to RFP (in red) and the GSK-3β biosensor (in green). In blue the electroporated cells, identified by H2B-BFP. (**G**) Bar charts showing 89.4% of DML electroporated cells are GSK-3β biosensor and SNAI1-positive. (**H**) Confocal stacks of somites and adjacent neural tube electroporated as described in *Figure 4H*, only in the DML with the GSK-3β biosensor (in green) and H2B-BFP (in blue, left panels) or double-electroporated in the DML with the GSK-3β biosensor (in green) and H2B BFP (in blue) and in the neural tube with DLL1 under the control of a neural crest-specific promoter (right panels). (**I**) Bar charts showing the% of GSK-3β biosensor-positive cells in the control (26.4%, in white) or with DLL1 expressed in the neural crest (64.6%, in black). (**J**) Confocal stacks of somites 17 hr after electroporation of GFP as control or DN-GSK-3β and immunostained for MYF5 (red). (**K**) Bar charts showing the% of electroporated cells that have entered the primary myotome in the control (45.8%, in white) of with the DN-GSK-3β (61.2%, in black). (**L**) Bar charts showing the% of electroporated cells that are MYF5-positive in the control (23.1%, in

*Figure 7 continued on next page*

*Figure 7 continued*

white) of with the DN-GSK-3β (57.2%, in black). In each panel are indicated the antigens that were detected by immunostaining, with the exception of native BFP blue fluorescence. Abbreviation: EP: electroporation; NT: neural tube. Scale bars: 50 μm.

GSK-3β biosensor that was indistinguishable from that obtained after electroporation of NICD (*Figure 9C,D*). This suggests that the regulation of GSK-3β activity by CD4-NICD is not a transcriptional response to NOTCH signaling.

These results prompted us to test whether CD4-NICD expression would mimic the effects of NICD on SNAI1 and MYF5. Indeed, we observed that the electroporation of CD4-NICD led to i) a massive stabilization of SNAI1 protein comparable to NICD alone (*Figure 9E,F*) and ii) to a robust increase of MYF5 expression in DML cells, not significantly different from the NICD alone (*Figure 9G,H*). These data demonstrate that the activation of MYF5 in DML cells is the result of a cytoplasmic function of NICD that can be uncoupled from its role as a transcriptional co-activator in the nucleus.

## Discussion

The findings presented here uncover a unique signaling module, triggered by a signal presented by migrating neural crest cells to selected progenitors in the somites. This signal is transduced into two distinct, but interrelated outcomes i) a cell fate decision in the receiving cells that leads to their entry into the myogenic program and ii) an EMT that allows their translocation into the myotome (*Figure 10*). While the key players in this module are well characterized, the demonstration that they cooperate to integrate cell fate determination and epithelial plasticity is novel.

EMT is intimately associated with essential changes in cell specification in many developmental processes. For instance, the massive and rapid cell rearrangement that takes place during neural crest formation or gastrulation in vertebrates vertebrate are associated with the acquisition of novel features that are characteristic of neural crest cells or of the forming mesoderm and endoderm, respectively (*Barrallo-Gimeno and Nieto, 2005*; *Joubin and Stern, 1999*). Inhibiting EMT in these models profoundly interferes with the associated cell fate changes. A large spectrum of secreted factors belonging to many signaling pathways have been shown to activate EMT. However, these mechanisms are unlikely to trigger at the same time the cell fate changes associated with the cell adhesion changes (*Barrallo-Gimeno and Nieto, 2005*; *Lamouille et al., 2014*). A remarkable aspect of the signaling module described here is that it targets GSK-3β, the main regulator of SNAI1 stability. In addition, aiming at GSK-3β may protect the β-catenin released during the EMT process from degradation, allowing it to accumulate sufficiently to act as a transcriptional co-factor, thereby resulting in a quasi-automatic coupling of EMT and β-catenin-dependent cell fate changes. Given the commonality of the key players described here in this paper, it is plausible that the same signaling circuitry could couple EMT to cell fate decisions in a wide variety of epithelia in normal and pathological conditions.

At both ends of this signaling module, NICD and β-catenin display unexpected functions. The 'canonical' role of NICD is to act as a co-transcriptional activator together with RBPJ. However, cytosolic interactions of NICD with various molecular partners that result in cellular responses independent of RBPJ, have been described in in vitro experimental settings (reviewed in [*Andersen et al., 2014*; *Ayaz and Osborne, 2014*]), but also in *Drosophila* (*Le Gall et al., 2008*). Our data uncover a novel in vivo function of NOTCH1 that takes place in the cytosol and leads to GSK-3β inhibition. It was shown that the transcriptional activity of NICD is modified by its phosphorylation in vitro, probably by GSK-3β (*Espinosa et al., 2003*; *Foltz et al., 2002*; *Schweisguth, 2004*). It is therefore not completely surprising that GSK-3β and NICD could (directly or not) interact in the DML. What is remarkable is the fact that this leads to a decrease in the overall activity of GSK-3β that has important consequences for the cell. The mechanisms for this are unknown. A hypothesis is that the response we observed results from a titration of GSK-3β kinase activity by NICD that is amplified by the stabilization of SNAI, itself a substrate for this kinase.

While it is largely accepted that β-catenin participates in adhesion and signaling functions in a mutually exclusive manner, changes in the transcriptional activity of β-catenin have also been

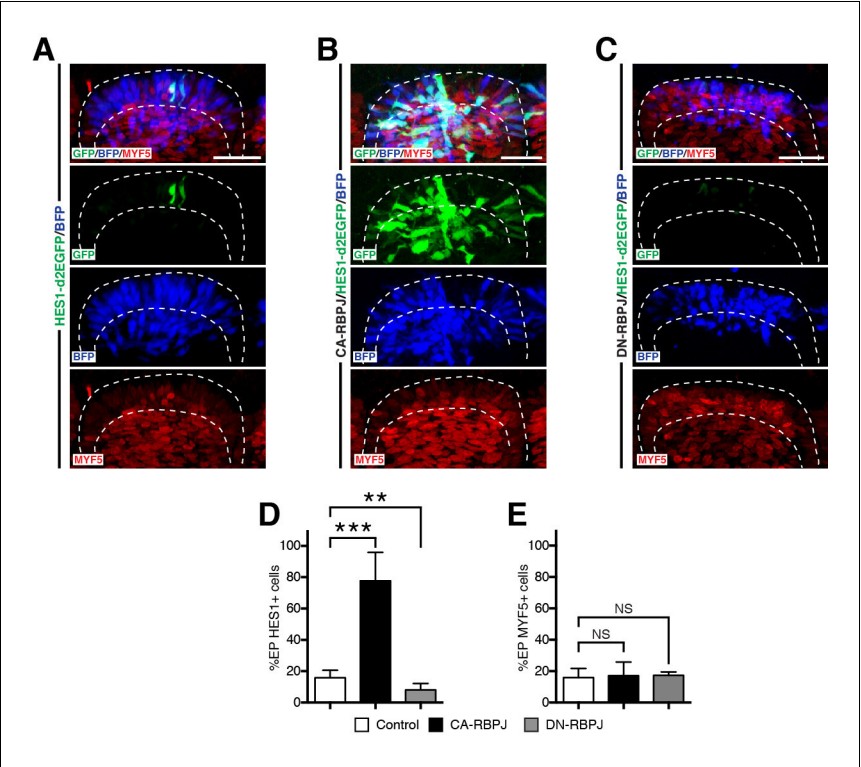

**Figure 8.** RBPJ does not regulate myogenesis in the DML. (A–C) Confocal stacks of somites 6 hr after co-electroporation of H2B-BFP (blue) and the NOTCH reporter (green) alone (A) or together with CA-RBPJ (B) or DN-RBPJ (C) and immunostained for MYF5 (red). (D) Bar charts showing the percentage of electroporated cells positive for the NOTCH reporter in the controls (15.8%, in white), with CA-RBPJ (77.6%, in black) or DN-RBPJ (8%, in grey). (E) Bar charts showing the percentage of electroporated cells positive for MYF5 in the controls (15.9%, in white), with CA-RBPJ (17.1%, in black) or DN-RBPJ (17.3%, in grey). In each panel are indicated the antigens that were detected by immunostaining, with the exception of native BFP blue fluorescence. Abbreviation: EP: electroporation. Scale bars: 50 μm.

associated with the disassembly of the junctional complex at the plasma membrane in vitro (*Brembeck et al., 2006*; *Gavard and Mège, 2012*; *Nelson and Nusse, 2004*). The results reported here provide direct evidence that this is also the case in vivo and demonstrate that the pool of β-catenin accumulated at the membrane and/or at the AJ has important signaling functions in addition to its established role in cell-cell adhesion.

Remarkably, we observed distinct thresholds of TCF/β-catenin transcriptional activity associated with discrete cellular outputs. It is known that WNT(s) expressed by the dorsal neural tube and transported by neural crest cells regulate the expression of WNT11 in the DML through canonical WNT signaling. WNT11 itself plays an essential role in the oriented elongation of early muscle fibers through planar cell polarity signaling (*Gros et al., 2009*; *Marcelle et al., 1997*; *Serralbo and Marcelle, 2014*). It is therefore not surprising to observe a TOPflash activity in DML cells that results from WNT ligand binding. The data presented here suggest that the WNTs carried by neural crest cells induce a low TOPflash response that regulates myotome's organization, but has no influence on MYF5 expression. On the contrary, DLL1, *via* the signaling module uncovered here, provokes the release of junctional β-catenin that leads to a high TOPflash response sufficient to trigger myogenesis. The reason for such different cellular responses is only speculative at present. It could be due to varying quantities of β-catenin entering the nucleus, low when WNT is presented by neural crest cells, high when it is (presumably massively) mobilized from the cell membrane. WNT11 and MYF5 would then be morphogen-like responses of DML cells to varying quantities of β-catenin. An alternative hypothesis is that of a qualitative difference in the transcriptional activity of β-catenin when it results from *bona fide* WNT signaling or when it is mobilized from the cell membrane. Rhee and

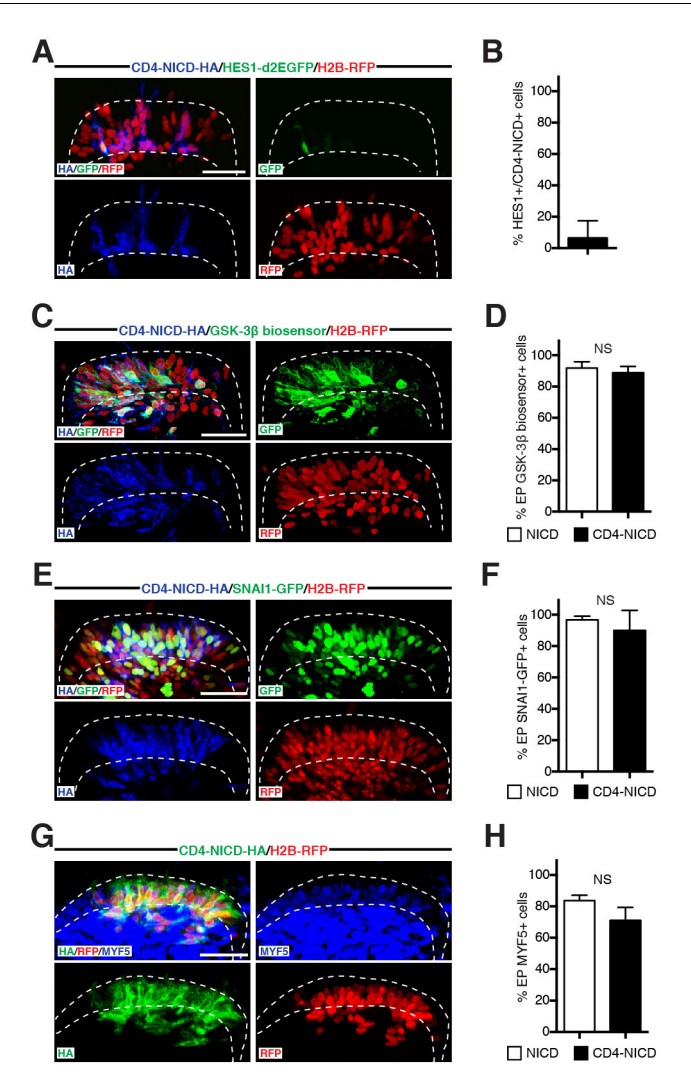

**Figure 9.** NOTCH1 controls myogenesis independently of its transcriptional role in the nucleus. (**A**) Confocal stacks of somites, 6 hr after co-electroporation of a HA-tagged, membrane-tethered NICD, CD4-NICD (in blue), a NOTCH reporter (in green) and H2B-RFP (in red). (**B**) Bar charts showing 6.4% of NOTCH reporter-positive cells electroporated with CD4-NICD. (**C**) Confocal stacks, 6 hr after electroporation of CD4-NICD (in blue), GSK-3β biosensor (in green) and H2B-RFP (in red). (**D**) Bar charts showing the percentage of GSK-3β biosensor-positive cells after electroporation with NICD (91.6%, in white) or with CD4-NICD (88.6%, in black). (**E**) Confocal stacks, 6 hr after co-electroporation of CD4-NICD (in blue), SNA1-GFP (in green) and H2B-RFP (in red). (**F**) Bar charts showing the percentage of SNAI1-GFP-positive cells after electroporation with NICD (96.7%, in white) or with CD4-NICD (90%, in black). (**G**) Confocal stacks, 6 hr after co-electroporation of CD4-NICD (in green), H2B-RFP (in red) and immunostained for MYF5 (in blue). (**H**) Bar charts showing the percentage of MYF5-positive cells after electroporation with NICD (83.7%, in white) or with CD4-NICD (71.1%, in black). In each panel are indicated the antigens that were detected by immunostaining. Abbreviation: EP: electroporation. Scale bars: 50 μm

colleagues have shown that β-catenin phosphorylated at the AJ on Y489 is transported into the nucleus and is transcriptionally active in vitro (*Rhee et al., 2007*). It is thus possible that post-translational phosphorylation of β-catenin potentiates or modifies its transcriptional activity, allowing a sharp separation between the WNT-dependent and the NOTCH-dependent cellular responses. This hypothesis is attractive, as it would explain why we did not observe any change in MYF5 expression even when WNT1 was over-expressed in neural crest cells and would also explain why in the

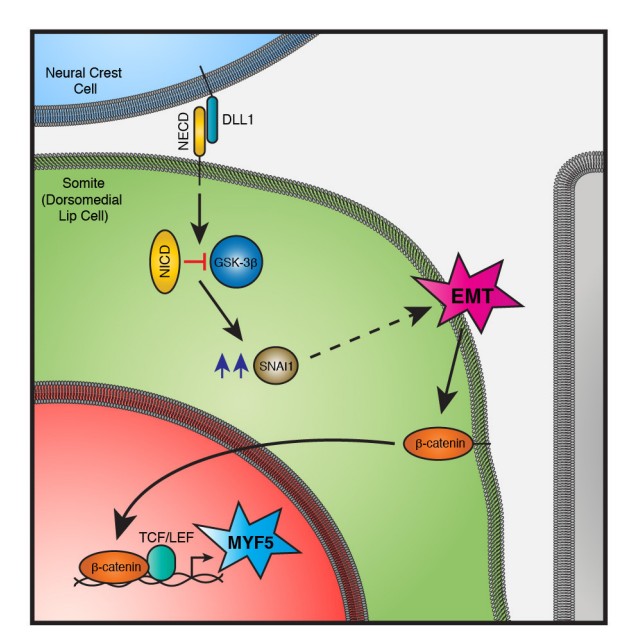

**Figure 10.** A signaling module in the DML that combines EMT and cell fate change. The physical contact with DLL1-positive neural crest cells triggers the cleavage of NOTCH receptor and the release of NICD in DML cells. NICD represses the activity of GSK-3β, independent of its transcriptional activity. The accumulation of SNAI1 (blue arrows) allows its translocation in the nucleus (not shown) where it activates EMT. This releases β-catenin from the adherens junction and/or membrane, leading it to enter into the nucleus where, together with TCF/LEF co-activator, it activates MYF5 expression. Abbreviations: DLL1: Delta-like 1; NECD: NOTCH extracellular domain; NICD: NOTCH intracellular domain; EMT: Epithelial to Mesenchymal Transition.

presence of the mutant Y489F β-catenin, endogenous β-catenin did not rescue MYF5 expression when NICD was over-expressed.

Finally, our study also provides a model that reconciles apparently divergent observations on the respective role of NOTCH and WNT signaling in the initial phases of myogenesis. The necessary function of NOTCH in this process (*Rios et al., 2011*) is confirmed, but the present work demonstrates that the NOTCH signal is a permissive one, while the instructive role is executed by β-catenin and TCF/LEF. This is in accordance with the role assigned to WNT signaling in somite myogenesis years ago (*Borello et al., 2006*; *Gros et al., 2009*; *Munsterberg et al., 1995*; *Stern et al., 1995*; *Tajbakhsh et al., 1998*); however, the results presented here shed an entirely novel light on the origin of the signal and the pathway that mediates it. More generally, the increasing complexity of the cellular responses triggered by migrating neural crest cells in somites is puzzling as it raises the question of how two distinct tissues have become so perfectly coordinated during evolution to generate such sophisticated interactions.

## Materials and methods

### Electroporation

The somite electroporation technique that was used throughout this study has been described elsewhere (*Gros et al., 2004*; *2009*; *Rios et al., 2010*; *2011*). Briefly, we targeted the expression of various constructs to the dorso-medial portion of newly formed interlimb somites of Hamburger–Hamilton (HH [*Hamburger and Hamilton, 1992*]) stage 15–16 chick embryos (24–28 somite). We have previously shown that this technique allows the specific expression of cDNA constructs in the DML, and that fluorescent reporters (e.g. GFP) are detected 3 hr after electroporation in this structure (*Gros et al., 2004*). Since we have analyzed most embryos 6 hr after electroporation, this implies that the molecules under study here have been acting during a narrow timeframe (about

3 hr). To target the neural crest cell population, we electroporated the dorsal neural tube of HH stage 13–14 chick embryos at the level of the pre-somitic mesoderm (*Rios et al., 2011*).

## Expression constructs

*The following constructs have been previously published:*

The WNT reporter 12Tf-d2EGFP contains 12 TCF/LEF1 binding sites (*Korinek et al., 1997*) upstream of a TK minimal promoter driving a destabilized GFP (d2EGFP, 2 hr half-life, Clontech [*Rios et al., 2010*]).

The CAGGS-H2B–RFP (provided by Dr. S. Tajbakhsh [*Rios et al., 2011*]) contains a fusion of Histone 2B with RFP, downstream of the CAGGS strong ubiquitous promoter (CMV/chick β-actin promoter/enhancer).

The CAGGS–EGFP (*Rios et al., 2011*) contains the CAGGS promoter followed by the EGFP reporter gene.

The constitutively active form of LEF1, CA-LEF1 (provided by Dr. A. Münsterberg) is described in *Abu-Elmagd et al. (2010)*.

The CAGGS-DN-MAML1–EGFP contains a truncated, dominant-negative form of the human Mastermind (DN-MAML1), fused with EGFP (*Weng et al., 2003*), downstream of the CAGGS promoter.

pCS2+ DKK1-flag (Addgene #16690) contains a human *DKK1* fused to a flag tag downstream of a CMV promoter (*Semënov et al., 2001*).

U3-DLL1 (*Rios et al., 2011*) was made by inserting the U3 evolutionarily conserved Sox10 enhancer sequences (*Werner et al., 2007*) in the TK-EGFP plasmid (*Uchikawa et al., 2003*).

A Tet-on (Clontech, Mountain View, CA) inducible system was used to activate NICD expression (*Rios et al., 2011*). The doxycyclin inducible system is composed of two plasmids that are co-electroporated: first, the pCIRX-rtTA (provided by O. Pourquié) contains a Tet-On Advanced transactivator (rtTA, Clontech) downstream of the CAGGS promoter. Second, the pBI-HA-NICD is the response plasmid (Clontech) in which the HA-tagged constitutively active form of the NOTCH1 receptor from chick, NICD (*Daudet and Lewis, 2005*) (provided by Dr. N. Daudet), was cloned.

CAGGS-cSNAI1 contains the wild type chicken Snail1 gene downstream of the CAAGS promoter upstream of an IRES EGFP (*Delfini et al., 2009*).

The dominant negative form of SNAI1 consists of the chicken SNAI1 in which the repressor domain was replaced by the VP16 activator domain of the Herpes simplex virus (*Delfini et al., 2009*).

SNAI1-GFP (Addgene #16225) contains a CMV promoter upstream of a human wild type Snail1 fused to EGFP (*Zhou et al., 2004*).

CAGGS NICD contains the chicken NICD downstream of the CAGGS promoter (*Rios et al., 2011*).

The GSK3 biosensor (pCS2 GFP-GSK3-MAPK, Addgene #29689) contains a CMV promoter upstream of a GFP molecule followed by a polypeptide tail that contains 3 GSK-3β phosphorylation sites, a priming site for MAPK/Erk and a site for the binding of E3 polyubiquitin ligases (*Taelman et al., 2010*).

HES1–d2EGFP (provided by Dr. R. Kageyama) contains the HES1 mouse promoter followed by a destabilized d2EGFP (*Ohtsuka et al., 2006*).

The constitutively active form of β-catenin was described in *Gros et al. (2009)*.

The dominant negative form of Dll1 has been described in *Rios et al. (2011)*.

A constitutively active form of RBPJ (pCMX-N/VP16-RBPJ, obtained from Dr. T. Honjo) comprises the CMV promoter driving a VP16 activation domain fused to the mouse RBPJ gene (*Kuroda et al., 1999*).

The dominant-negative GSK-3β (Addgene #29681) is a kinase dead variant of the Xenopus GSK-3β gene fused to an EGFP protein (*Taelman et al., 2010*).

The SNAI1 siRNA (as well as the Luciferase control) were described and tested in *Delfini et al. (2009)*. It leads to an efficient down-regulation of SNAI1 transcript level in chick tissues.

The NOTCH1 siRNA was described in *Rios et al. (2011)*. It leads to an efficient down-regulation of NOTCH1 transcript level in chick tissues.

*The following plasmids were constructed:*

HES1-DSRed-Express-DR was created by replacing the d2EGFP from HES1–d2EGFP (provided by Dr. R. Kageyama [*Ohtsuka et al., 2006*]) with a fast-folding, unstable variant of DSRed (Clontech).

The wild-type human β-catenin was amplified from pFG8 (provided by Dr. N. Plachta), then cloned into pCX-Myc (pCAGGS with 6xMyc tag, obtained from Dr. X. Morin).

Human β-catenin containing the Y489F (Tyr to Phe [*Rhee et al., 2007*]) mutation was amplified from Addgene #24197, then cloned into pCX-Myc.

The dominant-negative version of the mouse Wnt1 (obtained from Randy Moon [*Hoppler et al., 1996*]) was inserted into the pCLGFPA (*Scaal et al., 2004*).

The wild-type version of WNT1 (obtained from Randy Moon) was inserted into pCAGGS IRES H2B-GFP (provided by Olivier Serralbo) upstream of the IRES element.

The membrane tagged NICD was constructed using Gibson assembly (NEB) with the signal peptide FGFR2 (from *Danio rerio*), the extracellular and transmembrane domain of human CD4 (both obtained from Dr. J Kaslin) followed by the HA tagged NICD previously described, cloned either into the bidirectional vector tetracycline responsive vector or into the pCX-Myc described above.

The H2B-BFP has been made by replacing the RFP from the CAGGS-H2B–RFP described above by a TagBFP (Evrogen).

The SNAI1-RFP has been made by replacing the GFP from SNAI1-GFP (Addgene #16225) by RFP (Clontech).

CAGGS-BFP has been made by cloning the TagBFP (Evrogen) into the pCAGGS vector.

## Immunohistochemistry

Whole-mount antibody staining were performed as described (*Gros et al., 2009*). The following antibodies were used: rabbit polyclonals directed against chick myogenic regulatory factor MYF5 (obtained from Dr. B. Paterson (*Manceau et al., 2008*); 1/200); and anti-RFP (Abcam #62341, 1/1000); chicken polyclonal antibody against EGFP (Abcam #13970, 1/1000). The neural-crest-specific monoclonal antibody HNK1 was provided by Dr. A. Eichmann. Mouse monoclonal antibodies against c-myc-tag (Abcam #32072, 1/200), HA-tag (CST, #2367S, 1/100) and β-catenin (BD biosciences, #610154) were also used.

## Doxycycline-mediated induction of NOTCH signaling

We have tested the role of β-catenin from the membrane and AJ using a mutant (Y489F) form of β-catenin and a doxycycline inducible (Tet-On system) NICD. The association of β-catenin with proteins partners (α-catenin and cadherin) at the membrane is regulated by its phosphorylation at highly conserved tyrosine residues 142, 489 and 654 (*Lilien and Balsamo, 2005*; *Palka-Hamblin et al., 2010*; *Rhee et al., 2007*). In the neural tissue, the phosphorylation of β-catenin on tyrosine 489 reduces its affinity for N-cadherin and a mutant form of the β-catenin where this tyrosine is mutated to a phenylalanine (Y489F) cannot be mobilized from the junctional complex (*Rhee et al., 2007*). In DML cells, where β-catenin is also associated with N-cadherin (*Gros et al., 2005*), we hypothesized that cells in which endogenous wild-type β-catenin was replaced by Y489F mutant β-catenin would not be able to activate myogenesis. Due to the fast turnover of the junctional complex (*Baum and Georgiou, 2011*), and the rapid degradation of cytoplasmic β-catenin by the APC/Axin destruction complex (*Clevers and Nusse, 2012*), we speculated that the forced expression of the Y489F mutant β-catenin would rapidly replace the endogenous wild type (WT) β-catenin with its mutant counterpart, thereby allowing us to test the role of this variant in the induction of myogenesis. We tested whether the massive up-regulation of MYF5 expression resulting from the constitutive activation of NOTCH signaling (*Rios et al., 2011*) could be inhibited in presence of the mutant Y489F β-catenin.

Ten hours after electroporation of pCIRX-rtTA and pBI-HA-NICD, doxycycline (300 μl of a 0.75 μg/ml solution) was added onto the embryos. We have shown that the response plasmid is completely silent before doxycycline addition, while it is strongly and rapidly activated 6 hr after doxycycline addition (*Rios et al., 2011*).

## Confocal analyses

Dorsal views of somites shown in all figures are projections of stacks of confocal images taken using a 4-channel Leica SP5 confocal microscope (Leica). Confocal stacks of images were visualized and analyzed with the Imaris software suite (Bitplane). Cell counting was performed using the cell counter plugin (Kurt De Vos, University of Sheffield) within the ImageJ software (*Schneider et al., 2012*).

## Quantifications and statistical analyses

Electroporation results in the transfection of a portion of the targeted cell population, which is variable from embryo to embryo. To precisely evaluate the phenotypes obtained after electroporation of cell-autonomously acting cDNA constructs, the number of positive cells was compared to the total number of electroporated cells, recognized by an internal fluorescent reporter construct. On average, more than 700 cells were counted per point and the corresponding quantifications are shown in all figures.

To determine the fluorescence intensity of electroporated cells (*Figure 2D,E*), the surface of electroporated cells were rendered manually with Imaris software. The mean intensity for each cell and each channel in three-dimensions was collected for statistical analyses.

Statistical analyses were performed using the GraphPad Prism software. Mann–Whitney non-parametric two-tail testing was applied to populations to determine the P values indicated in the figures. In each graph, columns correspond to the mean and error bars correspond to the standard deviation. ***p value $< 0.001$, extremely significant; **p value 0.001 to 0.01, very significant.

## Acknowledgements

We thank Monash Micro Imaging (MMI) for imaging support; and Drs N Rosenthal, J-L Bessereau and PD Currie for critical reading of the manuscript. This work was funded by grants from the National Health and Medical Research Council (NHMRC, Australia) and by the Programme Avenir Lyon St Etienne (PALSE, France) to CM. CM is a Senior Research Fellow of the NHMRC. The Australian Regenerative Medicine Institute is supported by grants from the State Government of Victoria and the Australian Government.

## Additional information

### Funding

| Funder | Grant reference number | Author |
| --- | --- | --- |
| National Health and Medical Research Council | APP1026696 | Daniel Sieiro<br>Anne C Rios<br>Claire E Hirst<br>Christophe Marcelle |

The funders had no role in study design, data collection and interpretation, or the decision to submit the work for publication.

### Author contributions

DS, ACR, CEH, Designed the experiments, Performed the experiments, Analysis and interpretation of data, Drafting or revising the article; CM, Designed the experiments, Prepared the manuscript, Acquisition of data, Analysis and interpretation of data

### Author ORCIDs

Christophe Marcelle, http://orcid.org/0000-0002-9612-7609

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
