## [Decision Letter]

Thank you for submitting your work entitled "Cytoplasmic NOTCH and membranal β-catenin link cell fate choice to EMT during myogenesis" for consideration by *eLife*. Your article has been reviewed by three peer reviewers, one of whom is a member of our Board of Reviewing Editors, and the evaluation has been overseen by Fiona Watt as the Senior Editor.

The reviewers have discussed the reviews with one another and the Reviewing Editor, Marianne Bronner, has drafted this decision to help you prepare a revised submission.

Summary:

This manuscript addresses an interesting developmental biology problem – how cell behavior and fate are coordinated during morphogenesis – using myogenesis of the developing chick somite as a model system. The authors first resolve a previous conundrum in the field to show how Notch and WNT signaling pathways are integrated during myogenesis in the development somite. Interestingly, their results show that Notch triggers release of β-catenin from adherens junctions to the nucleus to activate MYF5, in a WNT ligand independent manner. This in itself is an important result. Although exchange between membrane and nuclear β-catenin pools has been previously postulated, this study directly demonstrates this occurrence in vivo. Second, using a GSK-3β biosensor, they find that Notch ICD inactivates GSK-3β, which in turn stabilizes SNAI1, which promotes EMT and also leads to activation of MYF5 expression. Moreover, this activity of NICD is important in the cytosol and independent of transcription.

These results are very interesting and highly appropriate for *eLife*. However, the presentation of the manuscript, including the title, is confusing and will require rewriting and reorganization in order to present the results and conclusions in a more direct and easier to interpret manner. The following changes should be made.

Essential revisions:

1) The manuscript clearly shows that both Notch and WNT pathways are required and that the WNT pathway is downstream of Notch. However, there is no need to invoke WNT itself since only β-catenin is required to transcriptionally activate myogenic factors and indeed they conclude this is ligand independent.

2) It is clear that SNAI1 is downstream of Notch and that GSK-3 inhibition potentiates SNAI1 function as an EMT inducer. Once SNAI1 is activated and stabilized, DML cells undergo an EMT that is presumably the consequence of cadherin repression and dismantling of adherens junctions. This in turn leads to the release of β-catenin to the cytoplasm, which translocates to the nucleus to activate MYF5. This connection is implied but not stated. They could test this using the Y489F mutant in this experiment to see if NICD still leads to elevated SNAI1-GFP but no activation of MYF5 in the presence of the mutant. Although this experiment is non-essential given the huge amount of data already present in the manuscript, it might tie the stories together a bit better.

3) The membrane tethered Y489F mutant is expressed in cells that already have endogenous β-catenin. The authors need to explain better why this construct prevents endogenous β-catenin from translocating to the nucleus.

4) It is clear that Notch signaling does not seem to use its canonical RBPJ-mediated transcriptional pathway. The question that remains unsolved is what is the mechanism by which Notch represses GSK-3 activity, but following the principles of *eLife* this is may be a question for an independent study.

5) In the Discussion, the authors should address how their proposed model reconciles with various reports showing that WNT receptors or membrane bound WNTs (Farin et al., Nature, 2016) can promote myogenesis.

In summary, the relevant players here are not Notch and WNT as stated in the title but rather Notch, SNAI1 and β-catenin. Similarly, alluding to "membrane" β-catenin in the title and in the text is misleading, as it is release from adhesion complexes in the membrane that allows β-catenin to fulfill its role in this process. As such, it is the SNAI1-mediated EMT that releases β-catenin from the membrane. In other words, it is a process mediated by EMT and β-catenin but independent of WNT ligand and Notch-mediated transcription. It is a very good example of how EMT can influence WNT signalling in a WNT-independent manner, as already suggested by Stemmer et al., Oncogene 2008 and discussed in EMT reviews. As already mentioned, all this is compatible with the model in Figure 10, but the title, the Abstract and the text should be written so that the model is readily followed and easy to understand. In general, the manuscript would benefit from text-editing to make it more concise and to the point. The way it is presently written may make it difficult for the non-expert reader.

[Editors' note: further revisions were requested prior to acceptance, as described below.]

Thank you for submitting your article "Cytoplasmic NOTCH and membrane-derived β-catenin link cell fate choice to epithelial-mesenchymal transition during myogenesis." for consideration by *eLife*. Your article has been reviewed by two peer reviewers, and the evaluation has been overseen by a Reviewing Editor and Fiona Watt as the Senior Editor.

In the present version, the text is difficult to follow and to understand. It is essential that you spell out 'Epithelial-mesenchymal transition' in the title.

The clarification about how the Y489F β-catenin works in the presence of the endogenous wild type protein should be added to the manuscript rather than just being in the response letter.

It is mandatory to re-write the manuscript to make this study readable for the non-expert.

---

## [Author Response]

1) The manuscript clearly shows that both Notch and WNT pathways are required and that the WNT pathway is downstream of Notch. However, there is no need to invoke WNT itself since only β-catenin is required to transcriptionally activate myogenic factors and indeed they conclude this is ligand independent.

There is a consensus in the WNT field to name the pathway that is dependent on β-catenin-dependent transcription the "WNT/β-catenin-dependent" pathway (the name "canonical" WNT signaling has also been used in the past), to differentiate it from alternate pathways (i.e. the Planar cell Polarity and the Ca^2+^-dependent pathways). This is the nomenclature we had decided to follow for our manuscript, despite the fact that one of the main messages of our study is that β-catenin can act as a transcription factor independent of WNT ligand. We understand the confusion that may rise from mentioning WNT in the text and we have revised the text throughout to address the point raised by the reviewers.

2) It is clear that SNAI1 is downstream of Notch and that GSK-3 inhibition potentiates SNAI1 function as an EMT inducer. Once SNAI1 is activated and stabilized, DML cells undergo an EMT that is presumably the consequence of cadherin repression and dismantling of adherens junctions. This in turn leads to the release of β-catenin to the cytoplasm, which translocates to the nucleus to activate MYF5. This connection is implied but not stated. They could test this using the Y489F mutant in this experiment to see if NICD still leads to elevated SNAI1-GFP but no activation of MYF5 in the presence of the mutant. Although this experiment is non-essential given the huge amount of data already present in the manuscript, it might tie the stories together a bit better.

Ours and Giulio Cossu's laboratories have shown that the expression of DN forms of TCF or β–catenin lead to an arrest of MYF5 expression (Borello et al., 2006; Gros et al., 2009). Borello and colleagues have further shown that the transcription of MYF5 in the mouse DML is controlled by 5 TCF/LEF binding sites. Thus, that MYF5 expression in the DML is a direct consequence of the binding of TCF and β–catenin onto its regulatory sequences is known. We have clarified this point in the Introduction (third paragraph).

3) The membrane tethered Y489F mutant is expressed in cells that already have endogenous β-catenin. The authors need to explain better why this construct prevents endogenous β-catenin from translocating to the nucleus.

We have done this experiment a number of times with each time the same result: Y489F mutant acts (even in the presence of NICD) as a dominant-negative β–catenin molecule. Why this is the case is speculative. We had a long discussion on this very point with Hans Clevers a few weeks ago. His group published in 2009 a study showing that GSK3β is separated into two pools, a minor one (3-5%) that interacts with β–catenin within the APC complex and a major one that acts with its numerous other targets (e.g. SNAI; Ng et al. JBC, 2009). Coherent with this study, NICD would likely interact with the "free" GSK3β available in the cytoplasm, blocking its activity (and allowing SNA1 to accumulate). In contrast, the GSK3β located within the APC complex could be "immune" to interaction with NICD, being already associated with various proteins within the complex. Thus, when NICD is placed in an Y489F setting, WT β–catenin cannot accumulate enough to drive MYF5 since the APC complex is likely still active.

A second hypothesis (based on our observation that MYF5 expression is strictly associated with a high WNT response) is that the amount of β-catenin needed to activate MYF5 cannot be reached (within the time frame of our experiments) by stopping its cytosolic degradation, but instead only after a massive mobilization from the cell membrane.

Finally, it is possible that the transcriptional activity of the β-catenin released from the cell membrane is distinct from the one that escapes the APC complex. This is based on very preliminary observation that would need solid confirmation.

One or a combination of the hypotheses described above could explain the failure of NICD to rescue MYF5 expression in Y489F conditions. We are pursuing some of those hypotheses at present.

4) It is clear that Notch signaling does not seem to use its canonical RBPJ-mediated transcriptional pathway. The question that remains unsolved is what is the mechanism by which Notch represses GSK-3 activity, but following the principles of eLife this is may be a question for an independent study.

This is indeed an important point. We have obtained preliminary biochemical data showing that NICD interacts in a complex with GSK3β and Akt in vivo.

Those results are encouraging, as they clearly show that NICD and the membrane-tethered NICD (CD4-NICD) co-immunoprecipitate with GSK3β and Akt, but they need to be repeated and improved. While the regulatory role of Akt on the phosphorylation (and therefore the inactivation) of Gsk3β is known, the interaction of NOTCH with Akt was never published. However, while this observation brings us one step further towards an understanding of NOTCH function in the cytosol, it leaves additional questions unanswered: e.g. we have no indication on whether NOTCH and Akt or GSK3 directly or indirectly interact and what is the consequence of this interaction on the activity of NOTCH itself. We provide a figure of those results to the reviewers to satisfy their curiosity, but because they are preliminary and incomplete, we do not wish to include those results in the manuscript. Because the role of NOTCH we uncovered is potentially of importance in a number of cellular contexts, we think these crucial questions deserve to be addressed in an independent study.

5) In the Discussion, the authors should address how their proposed model reconciles with various reports showing that WNT receptors or membrane bound WNTs (Farin et al., Nature, 2016) can promote myogenesis.

In fact, we have recently studied how WNT is delivered from neural crest to somites (Serralbo et al., Development 2014). We showed that WNT does not freely diffuse from migrating neural crest cells, but it is presented to the receiving DML cells by the Heparan Sulfate Proteoglycans family member, Dally-Like-Protein/GPC4. We demonstrated that GPC4 is necessary for the transfer of WNT from neural crest to DML cells. Thus, our results are coherent with those of Farin and colleagues, and in fact it could be that HSPGs are implicated in the cell-cell transfer of WNT they describe in the intestinal crypt. However, we show in the present study that WNT from neural crest does not play a role in myogenesis and we believe integrating the WNT signal from crest into our final model would add an unnecessary layer of complexity to it.

In summary, the relevant players here are not Notch and WNT as stated in the title but rather Notch, SNAI1 and β-catenin. Similarly, alluding to "membrane" β-catenin in the title and in the text is misleading, as it is release from adhesion complexes in the membrane that allows β-catenin to fulfill its role in this process. As such, it is the SNAI1-mediated EMT that releases β-catenin from the membrane. In other words, it is a process mediated by EMT and β-catenin but independent of WNT ligand and Notch-mediated transcription. It is a very good example of how EMT can influence WNT signalling in a WNT-independent manner, as already suggested by Stemmer et al., Oncogene 2008 and discussed in EMT reviews. As already mentioned, all this is compatible with the model in Figure 10, but the title, the Abstract and the text should be written so that the model is readily followed and easy to understand.

We understand the comment of the reviewers and we think the terms "membrane-*derived* β-catenin" would better reflect the activity we uncovered in this study. We therefore propose the following title: "Cytoplasmic NOTCH and membrane-derived β-catenin link cell fate choice to EMT during myogenesis".

In general, the manuscript would benefit from text-editing to make it more concise and to the point. The way it is presently written may make it difficult for the non-expert reader.

In an effort to make the text easier to read, we have removed all numbers from the main text. They are now in the figure legends. We believe this considerably improves the flow of the story with the consequence that it is easier to follow.

[Editors' note: further revisions were requested prior to acceptance, as described below.]

In the present version, the text is difficult to follow and to understand. It is essential that you spell out 'Epithelial-mesenchymal transition' in the title.

The clarification about how the Y489F β-catenin works in the presence of the endogenous wild type protein should be added to the manuscript rather than just being in the response letter.

It is mandatory to re-write the manuscript to make this study readable for the non-expert.

We have gathered all comments made by the reviewers in the first review and have addressed each of them below. As a result, entire paragraphs have been re-written, new experiments added or results presented differently in an effort to make a clearer story.

Numbered comments by reviewers

1) "there is no need to invoke WNT itself since only β-catenin is required to transcriptionally activate myogenic factors and indeed they conclude this is ligand independent".

We agree with this comment: naming WNT could be misleading. To avoid any confusion, we have now changed most references to WNT ligands, as long as the text still made sense after those changes. For instance, we replaced the term "WNT response" by "TOPflash activity" or " TCF/β-catenin transcriptional activity"; we also replaced the term "WNT reporter" by "TOPflash reporter", etc. In this way, we believe that there should be no confusion for the readers as to what they are looking at.

2) "They could test this using the Y489F mutant in this experiment to see if NICD still leads to elevated SNAI1-GFP but no activation of MYF5 in the presence of the mutant".

We have tested every steps of the signaling module in every possible way. One can indeed think endlessly about new experiments, e.g. blocking GSK3 or NOTCH and rescuing MYF5 with Snail, LEF or β catenin; or as proposed here, blocking β catenin and verifying that all steps upstream are still active. As mentioned by the reviewers, those experiments are non-essential and importantly they do not provide further evidence for the validity of the signaling module we uncovered.

*3) "The membrane tethered Y489F mutant is expressed in cells that already have endogenous β-catenin. The authors need to explain better why this construct prevents endogenous β-catenin from translocating to the nucleus"*.

We included an explanation (written in the previous rebuttal) of why this is in the Discussion section, fifth paragraph).

4)"The question that remains unsolved is what is the mechanism by which Notch represses GSK-3 activity"

We have answered this point in the previous rebuttal. We have obtained preliminary data showing that NOTCH co-immunoprecipitates with GSK-3 and Akt. We do not know whether this is a direct or indirect interaction and we are testing at the moment which domain of NOTCH is implicated in this interaction.

5)"In the discussion, the authors should address how their proposed model reconciles with various reports showing that WNT receptors or membrane bound WNTs (Farin et al., Nature, 2016) can promote myogenesis".

The paper of Farin et al. does not study myogenesis. If the point that the reviewers want to make is the transport of WNT evoked in that paper, we have recently shown that WNT is not secreted from the neural tube, but rather transported to somites by the migrating neural crest cells. This point is now clearly stated in the first paragraph of the Discussion. But that is it carried rather than secreted does not change the outcome: we show experimentally that the WNT transported by NC does not regulate myogenesis, if we inhibit it (with DN WNT1), or increase it (with WNT1, this is a new experiment we added to Figure 4, see below).

Additional comments of reviewers:

In the summary of the review, reviewers mentioned that *"alluding to membrane β-catenin in the title and in the text is misleading (…), it is the SNAI1-mediated EMT that releases β-catenin from the membrane".*

The aim of a title is to attract the attention on what is novel in a paper so that potential readers go one step further, at least until the abstract where this is all clearly stated. That an EMT releases β-catenin from the membrane is not novel. Likewise, it is established that β-catenin triggers myogenesis. What is crucial here is that it is the β-catenin derived from the membrane pool, acting as a transcriptional co-factor, and not the cytoplasmic one that does the job. It is our view that this finding has to be in the title as well as the other key finding, the new function of NOTCH in the cytoplasm.

On a more fundamental note, we have experimentally demonstrated that every step of this linear module leads to the same outcome, an activation of MYF5. Such that according to one's personal interests or likings, a molecule (e.g. Δ, Notch, GSK3, Snail, β-catenin) or a process (e.g. neural crest migration, EMT) may seem more crucial than another to the mechanism we describe. The fact is, all of them are necessary, and this module works only if all of them are there at the right place and moment. From the comments, it seems that reviewers believe that Snail and EMT are the driving force of myogenesis. In fact, it is easy to understand from the module we describe that the direct activation of MYF5 by CA LEF or CA β-catenin short-circuits the need for Snail or an EMT. We have done those experiments (they are not shown in the paper), we expectedly observe strong MYF5 activation, but indeed with no propensity of cells to translocate into the myotome, thereby dissociating de facto Snail and the EMT from myogenesis. In fine, the only required molecular event in the module we uncovered is the direct activation of MYF5 expression by β-catenin and LEF. The steps upstream are merely means to this end.

"It is a very good example of how EMT can influence WNT signalling in a WNT-independent manner, as already suggested by Stemmer et al., Oncogene 2008"

The paper from Stemmer shows that Snail physically binds to β-catenin and stimulates its transcriptional activity, independent of a transcriptional regulation of cadherin. This is quite different from what we observe in our study and we are not aware that the mode of action of Snail on the transcriptional activity of membrane-derived β-catenin that we uncovered has ever been described.

"In general, the manuscript would benefit from text-editing to make it more concise and to the point. The way it is presently written may make it difficult for the non-expert reader".

In an effort to make this study more understandable, and as suggested by the reviewers and the editor, we have included the responses we had included in the previous rebuttal about the mechanism that may explain how the Y489 mutant form of β-catenin may work in our system (see also answer #3 above).

We have also added at various places along the text (Results) and in the Discussion section explanations on the identity of the low TOPflash MYF5-negative DML cells. We have known what these cells are for some time and we extensively published on them. They are cells that respond to WNT carried by neural crest to somites (as shown in Serralbo and Marcelle Development 2014) by activating WNT11 in the DML through canonical WNT signaling (as shown in Marcelle et al. Development 1997). WNT11 itself is a positional cue used by muscle cells to orient in the antero-posterior axis of the embryo through planar cell polarity signaling (Gros et al., Nature 2009). Although their presence in the DML alongside the cell population that interests us in the present manuscript makes things a little more complicated, we have not found a way to avoid characterizing them. However, we found that the part of the manuscript that described this population (Figure 2) was rather obscure in the previous version of the paper. Therefore, we have changed the graphs in Figure 2 by a new graph (Figure 2) and we have re-written the entire paragraph demonstrating the presence of two (low and high) TOPflash-positive populations in the DML. We believe their identity is now clearer.

We have also added another panel in Figure 4 which shows that over-expression of WNT in neural crest does not change MYF5 expression in somites. This completes and reinforces the opposite experiment we had shown in the previous version of the manuscript, where we showed that a dominant-negative WNT does not change MYF5 expression (now in Figure 4).

The Abstract has been modified for more clarity.

As explained in the previous rebuttal, we have removed all numbers from the main text. They are now in the figure legends. We have also fixed a few glitches there and there.

In conclusion, we are aware that the text is somewhat complex. Its complexity is due to the fact that we took the challenge to dissect an entire pathway, from beginning to end and to place it in its biological context. Every step of the pathway we uncovered was tested in numerous ways. We assayed about 30 distinct constructs and reporters in this study. There are a lot of experiments to describe. We wanted in this study to make a point that the chick system is amazingly powerful to test intricate questions with relative ease and in a short time. We believe we succeeded at this task. Contrasting with this apparent complexity, the end product is a simple signaling module that may have far reaching implications in various models. This is the take home message of the study that we believe most readers will remember.